# Dynamics of Active Fluorescent Units (AFU) and Water Activity (a_w_) Changes in Probiotic Products—Pilot Study

**DOI:** 10.3390/foods12214018

**Published:** 2023-11-03

**Authors:** Katarzyna Sielatycka, Joanna Śliwa-Dominiak, Martyna Radaczyńska, Wojciech Juzwa, Mariusz Kaczmarczyk, Wojciech Marlicz, Karolina Skonieczna-Żydecka, Igor Łoniewski

**Affiliations:** 1Institute of Biology, Faculty of Exact and Natural Sciences, University of Szczecin, ul. Felczaka 3c, 71-415 Szczecin, Poland; 2Sanprobi Sp. z.o.o. Sp.k., ul. Kurza Stopka 5C, 70-535 Szczecin, Poland; joanna.sliwa@sanprobi.pl (J.Ś.-D.); martyna.radaczynska@sanprobi.pl (M.R.); mariusz.kaczmarczyk@sanprobi.pl (M.K.); wojciech.marlicz@pum.edu.pl (W.M.); igor.loniewski@sanprobi.pl (I.Ł.); 3Depratment of Biotechnology and Food Microbiology, Poznan University of Life Sciences, ul. Wojska Polskiego 28, 60-627 Poznań, Poland; wojciech.juzwa@up.poznan.pl; 4Department of Gastroenterology, Pomeranian Medical University in Szczecin, 70-204 Szczecin, Poland; 5Department of Biochemical Science, Faculty of Health Sciences, Pomeranian Medical University in Szczecin, 70-204 Szczecin, Poland; karolina.skonieczna.zydecka@pum.edu.pl

**Keywords:** probiotics, flow cytometry, active fluorescent units, stability of probiotic products, bacterial enumeration

## Abstract

The flow cytometry method (FCM) is a widely renowned practice increasingly used to assess the microbial viability of probiotic products. Additionally, the measurement of water activity (a_w_) can be used to confirm the presence of viable cells in probiotic products throughout their shelf lives. The aim of this study was to investigate the correlation between changes in a_w_ and variations in active fluorescent units (AFU), a unit commonly used in flow cytometry method, during the aging of probiotic products containing freeze-dried bacteria. We controlled the stability of probiotic products for bacterial counts (using ISO 19344 method) and a_w_ levels in commercially available capsules containing freeze-dried bacteria such as *Lactobacillus* sp. or combinations of *Lactobacillus* sp. and *Bifidobacterium* sp. in standard conditions (25 ± 2 °C and 60% relative humidity) over a period of 24 months. During this time, the bacterial contents decreased by 0.12 Log_10_ in the single-strain product, by 0.16 Log_10_ in the two-strain product and by 0.26 Log_10_ in the multi-strain product. With the increase in a_w_, the number of bacteria decreased but the a_w_ at the end point of the stability study did not exceed 0.15 in each of the three tested products. FCM combined with a_w_ is a prospective analysis that can be used to assess the stability of probiotic products, both for its ability to detect bacterial viability and for practical (analysis time) and economic reasons.

## 1. Introduction

Over the years, probiotics, also known as “good bacteria”, have gained increased recognition because they have the potential to rebuild intestinal microflora. The probiotic industry is constantly developing, and companies use probiotics in cosmetics, food products, dietary supplements, etc. Several definitions of the term “probiotic” have been used over the years, but that of the World Health Organization is the most widely used and recognized: “live microorganisms that, when administered in adequate amounts, confer a health benefit on the host” [1].

Controlling and knowing the number of bacteria in probiotics is essential to ensure product quality for several reasons, including efficacy, safety, and regulatory compliance [2]. If there are too few bacteria, the probiotic may not have the intended health benefits. On the other hand, if there are too many bacteria, it may have a negative effect on the body, potentially leading to infections, especially in people with weakened immune systems [3,4]. Several countries’ regulations require that the minimum number of viable microorganisms at the end of the storage period be indicated on the label of the probiotic product [5]; therefore, controlling this number is necessary to meet regulatory requirements and ensure that the probiotic is safe for consumption [2]. In addition to controlling the number of bacteria, it is also important to monitor other parameters of probiotic products such as stability and resistance to environmental conditions. This ensures that the probiotics remain effective throughout storage and can reach the gut alive.

We have observed that stability testing is of interest to probiotic manufacturers as they recognize the large number of factors that can affect the integrity and stability of probiotics and ultimately their efficacy. Stability testing of finished products is necessary to determine the expected viability of the bacteria and to show how long the product will remain stable under appropriate storage conditions. Very important factors relevant to these studies are the time and cost of performing the analysis, as well as useful methods to assess the function of the probiotics. One type of proposed stability test is a real-time stability test, which consists of monitoring the amount of bacteria at specific time points in combination with the monitoring of a_w_. We have found this method to be an accurate, simple, rapid and cost-effective way to determine the shelf life of a dietary supplement.

The traditional, and one of the oldest and most common, methods used to assess the total amount of microorganisms in a probiotic product is the plate count (PC) method, which evaluates bacterial cells’ ability to proliferate into detectable colonies on agar media and specifies the results as colony-forming units (CFU). However, this method is laborious, requiring specialized equipment for the incubation of bacteria under anaerobic conditions and many days to obtain results. For this reason, a PC method for real-time control of the microbial count in the production process is not the best solution because in the case of a large number of samples, this method is expensive. Moreover, we can use that method only to detect cultivable bacteria but not the whole population of live bacteria.

Methods for assessing cell viability are widely described in the literature [6,7,8,9], and scientists are looking for new cheaper and faster techniques to quantify the amount of microorganisms in the final product [10]. In order to shorten the time needed to obtain results and obtain more reliable data on bacterial viability, a more precise method was used for counting viable microorganisms—flow cytometry method (FCM). Using FCM and special dyes, live, damaged and dead cells can be labeled, allowing a qualitative and quantitative assessment of the microorganism proportion in the probiotic formula. This is also crucial for identifying microorganisms that survive the production process but are not viable. The principle of FCM is that cells are sent through a nozzle and analyzed one by one by a laser. The forward light scatter provides information about the cell and the side scatter provides information about its granularity and morphology. Instead of CFU, FCM produces results as AFU (active fluorescent units), which represent the viable (intact) cells, or n-AFU (non–active fluorescent units), which represent non-viable cells. The published international standard ISO 19344 (IDF 232) clearly provides a method for the quantification of lactic acid bacteria in fermented products, starter cultures and probiotics used in dairy products via flow cytometry (FC) [11]. This is the result of joint research by ISO and the International Dairy Federation (IDF), which is approved for the analysis of freeze-dried products.

A crucial parameter affecting the stability of probiotics is water activity (a_w_) because determining a_w_ data during the formulation, manufacturing and storage process is a great method of increasing confidence so that sufficient levels of viable cells are available throughout its shelf life [12,13]. Having regard to direct enumeration can show how many viable cells are left at the time of analysis, a_w_ measurement can assist in predicting the death rate of viable cells over a period of time, and if any issues occurred during the manufacturing process that were not picked up by the enumeration testing. The parameter of a_w_, despite its great importance in the process of assessing the stability of probiotics, is not always taken into account [14,15,16]. Of note, a_w_ is described in United States Pharmacopeia (USP) and recommended to use in testing of raw materials and pharmaceutical products’ stability [17].

Taking into account that FCM is a method increasingly used to control the quality and stability of probiotic products, further research using this method is still needed and therefore, one such possibility is to show the correlation between FCM and a_w_, which is the main factor responsible for the stability of probiotics.

This study aims to investigate how changes in water activity correlate with variations in active fluorescent units during the aging of probiotic products containing freeze-dried bacteria.

## 2. Materials and Methods

### 2.1. Samples

The samples used in this study contained lyophilized bacterial cultures enclosed in a capsule. The stability of cultures was assessed in a single-strain product with *Lactobacillus plantarum* 299v, a two-strain product containing *Lactobacillus helveticus* Rosell-52 and *Bifidobacterium longum* Rosell-175, and a four-strain product containing: *Bifidobacterium bifidum* Rosell^®^-71, *Lactobacillus helveticus* Rosell^®^-52, *Lactococcus lactis* Rosell^®^-1058 and *Lactobacillus casei* Rosell^®^-215. Sanprobi (Poland) final probiotic products were used for the research.

### 2.2. Storage Conditions of Samples

Samples of product were stored in blisters composed of pharmaceutical aluminum foil/Pentapharm^®^ Aclar^®^ S03 (PVC/ACLAR/PVC) with a water vapor transmission rate (WVTR) of 0.11 g/m^2^ × d at climate zone II (25 °C ± 2 °C and 60% relative humidity) for 2 years. After regular time intervals, the numbers of live and dead cells were measured. The storage conditions were set based on the information contained in the International Council for Harmonization of Technical Requirements for Pharmaceuticals for Human Use (ICH).

### 2.3. Sample Preparation for Flow Cytometry

For flow cytometry analysis and viable count estimation, 10 ± 2 g of lyophilized bacteria, poured out of the finished capsules, was suspended in 90 mL of buffered peptone water (Oxoid). To dissolve and homogenize the test samples, a stomacher (BagMixer^®^, Interscience, Saint-Nom-la-Bretèche, France) was used. The samples were rehydrated for 15 min, and then, serial dilutions in sterile peptone water were created.

### 2.4. Flow Cytometry Staining Procedure

Cell staining was performed according to ISO 19344: IDF 232 (2015). 100 µL of the appropriate dilution sample was added to 880 µL of sterile PBS (Invitrogen, USA). Next, 10 μL of SYTO24 (5 mM) green-fluorescent nucleic acid stain (Invitrogen, USA) and 10 μL of propidium iodide (PI) (0.2 mmol/L) (Invitrogen, USA) were added to the staining tube, vortexed and incubated for 30 min in 37 °C.

### 2.5. Flow Cytometry Settings

Samples were analyzed using a CytoFLEX LX (Beckman Coulter, Indianapolis, IN, USA) flow cytometer and the CytExpert (Beckman Coulter, USA) data analysis software. Before each analysis, the instrument condition was checked using CytoFLEX Daily QC Fluorospheres (Beckman Coulter, USA). Forward scatter (FSC), side scatter (SSC) and two fluorescence signals were measured. A 488 nm laser with 50 mW power was used for the excitation of both used fluorochromes. SYTO24 fluorescence was recorded using a 525 nm longpass filter and bandpass filter with a transmission at 525/40 nm. PI fluorescence was recorded using a 690 nm longpass filter and bandpass filter with a 690/50 nm transmission. To minimize the noise level, a threshold was set on the FITC channel at 609. The sample flow rate was set to 60 μL per minute, and each time, 100 μL of sample was recorded for subsequent calculations.

### 2.6. Gating Strategy and Calculation

The gating strategy and calculation method, which is used to ensure the real-time stability of probiotics, was previously described by our team in 2021 and is shown in Figure 1 [18]. The FSC vs. SSC dot plot shows all bacteria and then distinguishes between live and dead cells as shown in the SYTO 24 vs. PI dot plot. Due to the size of the bacteria, the analyses were performed on a logarithmic scale. FCM results were expressed as AFU/g, n-AFU/g and TFU/g. AFU cells contain CFU and cells in a viable but non-culturable state (VBNC) that retain an intact cell membrane and/or demonstrate sublethal cell membrane damage characterized by the limited penetration of propidium iodide. TFU represents the number of live (AFU) and dead (n-AFU) cells. By analyzing the number of probiotic bacteria defined as AFU/n-AFU and TFU, we were able to monitor the viability of bacterial cells during the storage of the final product using stability tests. To be sure of the quality of the products, three series of each product were subject to stability tests each year.

### 2.7. Determination of Water Activity

The water activity of the analyzed products was assessed according to ISO 21807: 2004 “Microbiology of food and animal feedstuff”, a determination of water activity that provides the basic principles and requirements of physical methods to determine the a_w_ of products intended for human consumption and the feeding of animals. Water activity can be used to predict microbial growth and determine the microbiological stability of a food product, as well as providing an important, quantitatively determinable criterion for estimating how long a foodstuff can be kept. Briefly, 8 capsules of the final product were randomly taken from a specific production batch and placed in a plastic measuring pan. The test was carried out using the ROTRONIC HC2-AW-USB probe for a_w_ measurement. The measurement time for all products was 30 min and was determined in the method validation process at Sanprobi Research and Development Center.

### 2.8. Microbiological Examination of Studied Products

The microbiological quality and purity of studied products were assessed in accordance with the criteria laid down in European Pharmacopoeia (E.P.) 10.1. (chapter 2.6.12, 2.6.13) and evaluated at the beginning (0) and at the 12th and 24th months of the stability test. European Pharmacopoeia criteria concern microbial purity, which is very important for the safety of dietary supplements as non-sterile biological products. According to these criteria, the product should be free of pathogens. The following microbial assays were performed: total aerobic microbial count (TAMC) and total mold and yeast count (TYMC) as microbial enumeration tests; presence and count of bile-tolerant Gram-negative bacteria (*Enterobacteriaceae*); and presence of *Eschericha (E.) coli*, *Staphylococcus (S.) aureus* and *Salmonella* sp. for qualitative examination tests for specified microorganisms. The methods and media described in E.P. were used with some modifications. The TAMC count was performed by plating 1 mL of decimal dilutions on count agar sugar-free FIL-IDF (Millipore). Plates were incubated at 35 °C for 3–5 days. The results are presented as colony-forming units per g (CFU/g) (Table 1, Table 2 and Table 3). The TYMC count was performed by plating 1 mL of decimal dilutions on Sabouraud dextrose agar (Oxoid). Plates were incubated at 25 °C for 5–7 days. The results are presented as colony-forming units per g (CFU/g) (Table 1, Table 2 and Table 3).

For qualitative examination, 10 mL of prepared sample was added to 100 mL tryptic soy broth (Oxoid) and incubated at 35 °C for 24 h. The further procedure was dependent on the determination of the absence or limited occurrence of specified microorganisms that may be detected. *Enterobacteriaceae* was transmitted to enterobacteria enrichment broth, Mossel (Oxoid) (incubation for 24–48 h at 35 °C), and then transmitted to violet-red bile glucose agar (VWR) (incubation for 24–48 h at 35 °C). *E. coli* was transmitted to MacConkey broth (Oxoid) and incubated for 24 h at 44 °C, and then transmitted to MacConkey agar (Oxoid) and incubated for 24 h at 35 °C. *S. aureus* was transmitted on mannitol salt agar (Oxoid) and incubated for 24–72 h at 35 °C. *Salmonella* sp. was transmitted to Rappaport Vassiliadis broth (Oxoid) and incubated for 24 h at 35 °C, and then transmitted on xylose, deoxycholate agar (Oxoid) and incubated for 24 h at 35 °C. For the presence and count of bile-tolerant Gram-negative bacteria (*Enterobacteriaceae*), as well as the presence of *E. coli* and *S. aureus,* the results are presented as presence/absence per 1 g. For *Salmonella spp.* the results are presented as presence/absence per 10 g (Table 1, Table 2 and Table 3).

### 2.9. Statistical Analysis

A mixed-effect linear model was used to analyze changes in AFU (log_10_ transformed) and water activity during a two-year period. The model was designed as a random intercept, with time serving as a fixed effect and the probiotic product serving as a random effect. The model was fitted in R (https://cran.r-project.org/) (accessed on 10 June 2023) using the lme4 (1.1–31) package. The model’s output was graphically summarized using the predictor effect plots built using the effects package (version 4.2.-2) for R. Predictor effect plots can help explain the relationships between the predictors and the response as an alternative to tables of regression coefficients. *p* value < 0.05 was considered statistically significant.

## 3. Results

The trends of all tested product samples clearly indicate that a reduction in membrane integrity is observed during stability tests over a period of 24 months (Figure 2A–F). A flow cytometric evaluation of the viability of microbial cells, using SYTO24 to stain cellular nucleic acids and PI to evaluate the integrity of cellular membranes, enabled a basic differentiation between live and dead cells. The protocol improved the resolving power of single PI staining, and thus facilitated to some extent the detection of dormant and injured cells. This is demonstrated in Figure 1, where the separate subpopulation is visualized in between the sub-populations of live and dead microbial cells.

As shown in Figure 2A,B, 20.85% less bacteria in one capsule than at point “0” was noticed after two years of production, only containing *Lactobacillus plantarum* 299v. However, the result is still above the limit specification established in our laboratory, and the product is still qualified as safe and suitable for consumption. Decreasing the number of bacteria, assessed by the FCM, was 0.12 Log_10_ AFU/g (3.63 × 10^10^ AFU/g). Figure 2C,D shows the decrease in *Lactobacillus helveticus* Rosell-52 and *Bifidobacterium longum* Rosell-175 bacteria, and the difference is 0.16 Log_10_ AFU/g (2.81 × 10^10^ AFU/g). Multi-strain products, as shown in Figure 2E,F, characterize the greater difference in number of live and dead bacteria cells equal 0.26 log_10_ AFU/g (1.18 × 10^10^ AFU/g).

The microbiological quality and purity of studied products were assessed and are presented in Table 1, Table 2 and Table 3. The results are presented as colony-forming units per g of product (CFU/g) for quantitative examinations and as present/absent for qualitative tests. The analyzed product did not contain any Gram-negative bacteria (*Enterobacteriaceae*) that tolerate bile, *E. coli* and *S. aureus,* and the number of total aerobic microbial, yeast and molds count was at a low level. All results (Table 1, Table 2 and Table 3) are compliant with the European Pharmacopoeia (EP) criteria and none of the analyzed samples exceeded limits.

Water activity level for probiotic products should not exceed 0.2 [12]. For freeze-dried products, we defined that the a_w_ limit should be below 0.15, and all tested products meet this criterion. The lowest level of water activity is shown by the multi-strain product, as shown in Figure 3C. Evidently, in our study, a decrease in the number of AFU/g and an increase in a_w_ is observed (Figure 3A–C and Figure 4).

It is worth noting that, in each of our products, the amount of AFU decreases with the increase in a_w_, which never surpasses a value of 0.15 over a period of 24 months, as shown in Figure 3.

## 4. Discussion

In the pilot study we presented, our research team was the first to focus on a simultaneous combination of two methods: FCM, which is the preferred tool for assessing bacterial cell viability, and a_w_ level monitoring. The methods we used are complementary, as evidenced by the opposite and highly statistically significant changes in the measured parameters during probiotic storage. We have identified a methodological gap within the probiotic industry and believe that the development of a rapid and cost-effective approach to determining the shelf life of probiotic bacterial populations will yield valuable data.

In our study, we employed an innovative cytometric method, which is still relatively uncommon in the field of bacterial enumeration, and it is worth mentioning that it provides a rapid and accurate analysis of cell counts in the sample.

However, as with any method, there are limitations and potential sources of error that must be considered. Firstly, bacteria can form cell aggregates, which can affect the accuracy of the measurement. These aggregates can be counted as individual cells, resulting in an underestimate of the cell count. Secondly, if a low concentration of bacterial cells is used in the sample, it may be difficult to accurately assess the cell quantity because too few cells can lead to inaccurate results. Thirdly, the presence of contaminants in the sample, such as debris or other microorganisms, can lead to false results or interfere with the measurements. It is therefore important to clean the sample thoroughly before conducting the analysis. Fourthly, the technical error due to improper sample preparation, instrument calibration or other factors may affect the accuracy of the results. Therefore, it is necessary to perform appropriate control and calibration procedures to minimize this type of error. Despite these few limitations, FCM still has an advantage over PC methods, and using the researcher’s experience and familiarity with the instrument, we are able to produce reliable results.

The credibility of this method is often questioned; however, the recent publications by Visciglia et al. (2022) and Sielatycka et al. (2021) clearly indicate that the FCM represents a valid method and an innovative tool for studying bacterial populations [18,19].

In 2015, the ISO 19344 method was developed as a new viability method for probiotic cultures, providing an alternative approach to determine the viability of these cultures. However, one significant aspect seems to be lacking in this method, which is the inclusion of how to compare flow cytometry data with traditional plate count data. To address this limitation, further research and standardization efforts need to be undertaken to establish a framework for comparing FCM data with plate count data.

The second important parameter that affects the quality of the product is water activity. Too high a_w_ values mean a significantly increased risk of the development of pathogenic microorganisms and the presence of bacterial toxins in the product, which has a direct impact on the safety of consumers using it.

Vesterlund et al. (2012) showed that for three probiotics containing the *Lactobacillus rhamnosus* GG strain with a_w_ values of: 0.43; 0.22 and 0.11, the most microbiologically stable during 14 months of storage at 22 °C turned out to be the preparation with the lowest a_w_ value [12]. For the growth of most Gram-negative bacteria, the a_w_ value must be at least 0.97, and for the growth of molds, it must be at least 0.80. However, at an a_w_ of 0.60, osmophilic yeasts and xerophilic molds can grow.

Martins et al. (2019) attempted to determine the optimal values of water activity and powder temperature required for the production of long-life dry matter of *Lactococcus lactis* spp. They showed that to obtain a stable product after drying, a_w_ should be 0.198 at 52 °C [16].

Not only at the production stage of probiotics, but also during their storage, a_w_ plays an important role. Weinbreck et al. (2010) showed that for the probiotic containing the *Lactobacillus rhamnosus* GG strain, the a_w_ value during storage was of great importance in the context of bacterial cell survival. Comparing the probiotic powder stored at a_w_ of 0.70 and 0.15, the probiotic stored at a_w_ of 0.70 was characterized by significantly lower bacterial survival [20].

Water activity plays an important role in both the production and storage of probiotics, and the lower its level, the greater the microbiological stability of the product. Therefore, the increase in a_w_ level can be treated as a predictive factor for the destruction of probiotic bacteria, which is independent of the number of probiotic strains in the product. It should be emphasized that even the use of a blister with a very low WVTR does not completely protect the product from a_w_ growth; however, it does ensure its proper quality.

Improving the bacterial enumeration method is very important because it requires further optimization in the next-generation probiotics industry [2]. This is certainly a new research direction that we are following. Non-cultured, heat-killed, tyndallized or micronized microorganisms [21,22] are being increasingly reported in the literature, and “factors of probiotic origin” [23] may also benefit health. These bacteria can be referred to as postbiotics. In addition, in recent years, there has been a growing interest in “new” probiotics belonging to intestinal microbes. Most of the promising new strains are strictly anaerobic strains (e.g., *Akkermansia municiphila*, *Faecalibacterium prausnitzii*, *Eubacterium hallii*, etc.), where technological difficulties related to their industrial production are observed and the industry faces a challenge in their standardization. Classical microbiology is difficult to implement with these new species due to their often-unknown growth requirements, lack of a counting methodology and the need to work in anaerobic laboratories [24]. In some extreme situations, certain strains of microorganisms require the presence of other strains for growth. Therefore, by nature, they cannot be isolated and cultured in vitro, and thus, FCM can be used in the qualitative and quantitative assessment of these probiotics. Combination of FCM and a_w_ is essential in the process of assessing the stability of postbiotics, as evidenced by the specification of pasteurized *Akkermansia muciniphila* MucT approved in the European Food Safety Authority (EFSA) opinion concerning safety of this product [25]. Another possible application of FCM is the assessment of the growth phase of bacteria [26], which affects their immunological properties and can be used to produce targeted and personalized probiotics.

The most important limitation of the study is that the methods we used do not allow for the identification of individual strains in the tested samples. However, so far, no cytometric method has been developed and accepted by the International Organization for Standardization or Pharmacopoeia that could be used for the identification of individual strains. The problem of analyzing the viability of individual strains in multi-strain probiotics is still very relevant, but it has not been solved yet and requires further research [10].

Therefore, we would like to emphasize that the results presented here are of a pilot nature and have the potential to be used in preparation of a new method for assessing the quality of probiotics in the future. However, we are aware that we need many more results, method validation and interlaboratory comparisons to achieve this goal.

In summary, FCM in combination with a_w_ can be prospective methods used for assessing the stability of probiotic products, both in their ability to detect bacterial viability but also for practical (analysis time) and economic reasons. It is worth monitoring the relationship between a_w_ and AFU during the production stage, which can save time and money and affect efficiency. There are many possibilities to use such a research scheme; however, further research into using the results in probiotics stability testing is warranted.

## Figures and Tables

**Figure 1 foods-12-04018-f001:**
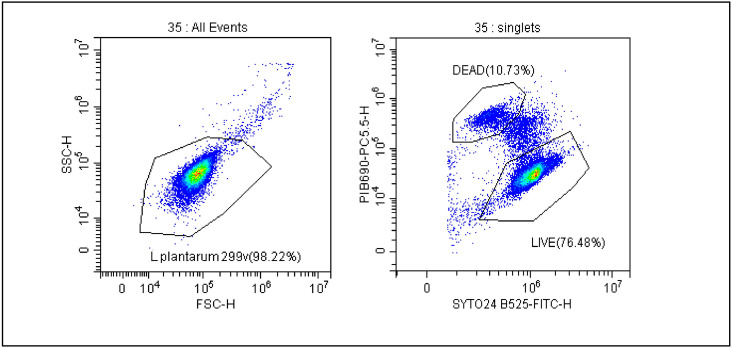
Gating strategy for evaluation of the number of AFU (active fluorescent units) and n-AFU (non-active fluorescent units) cells. Cells were displayed in forward scatter (FSC) vs. side scatter (SSC) plot in logarithmic scale, and a gate named *L. plantarum* 299 v is used to capture the cells of interest. Right dot plot shows gating strategy for live and dead *L. plantarum* 299 v analysis where SYTO24 and PI were employed.

**Figure 2 foods-12-04018-f002:**
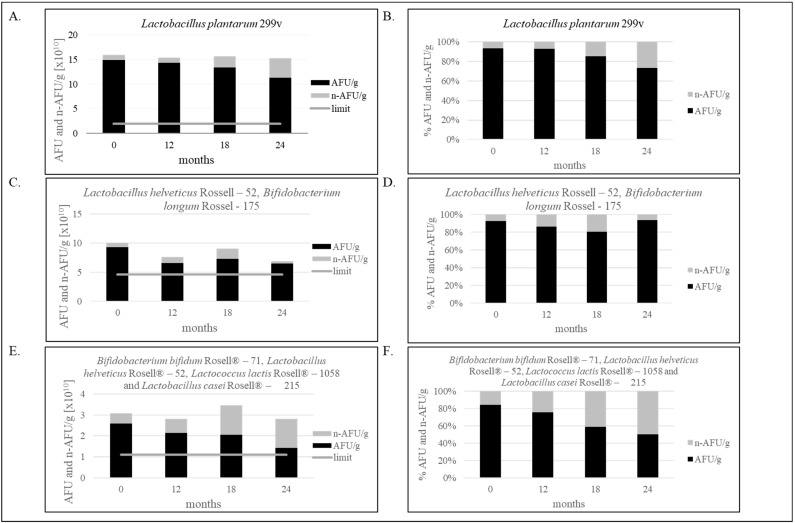
(**A**–**F**). Effect of storage condition at 25 °C on bacterial stability. (**A**,**B**)—The stability of the single-strain product containing *Lactobacillus plantarum* 299v for 24 months; (**C**,**D**)—The stability of the two-strain products containing *Lactobacillus helveticus* Rossell-52, *Bifidobacterium longum* Rossel-175 for 24 months; (**E**,**F**)—The stability of the multi-strain product containing *Bifidobacterium bifidum* Rosell^®^-71, *Lactobacillus helveticus* Rosell^®^-52, *Lactococcus lactis* Rosell^®^-1058 and *Lactobacillus casei* Rosell^®^-215 for 24 months. Panels (**A**,**C**,**E**) show the ratio of AFU to n-AFU changing during 24 months in one gram of product. Panels (**B**,**D**,**F**) show the percentage of AFU and n-AFU cells in the total population of bacteria defined as TFU (total fluorescent units) in one gram of product. Flow cytometry method (FCM) and evaluation of water activity were verified in our laboratory. Measurement uncertainty has been estimated in accordance with ISO 19036 and is based on a standard uncertainty multiplied by a coverage factor of k = 2, providing a level of confidence of approximately 95 % U = k × uc, (k = 2, *p* = 95 %). Combined standard uncertainty has been taken as equal to the intralaboratory reproducibility standard deviation (Sr = 0.181). a_w_: U = 0.016a_w_; FCM: the uncertainty for *Lactobacillus plantarum* 299v: U = 0.35 log_10_ cfu/g; the uncertainty for *Lactobacillus helveticus* Rosell-52 and *Bifidobacterium longum* Rosell-175: U = 0.23 log_10_ cfu/g; the uncertainty for multi-strain products containing *Bifidobacterium bifidum* Rosell^®^-71, *Lactobacillus helveticus* Rosell^®^-52, *Lactococcus lactis* Rosell^®^-1058 and *Lactobacillus casei* Rosell^®^-215: U = 0.36 log_10_ cfu/g. The figure is representative of one production series.

**Figure 3 foods-12-04018-f003:**
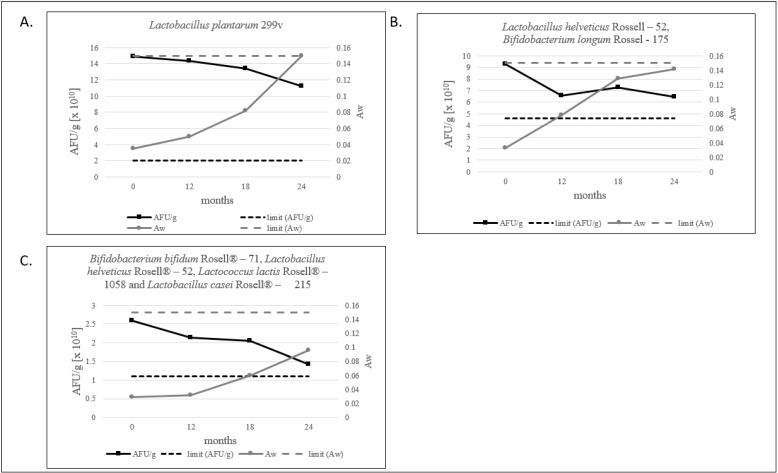
(**A**–**C**). The ratio of the AFU per 1 g of product and a_w_ value measured during 24 months of stability testing.

**Figure 4 foods-12-04018-f004:**
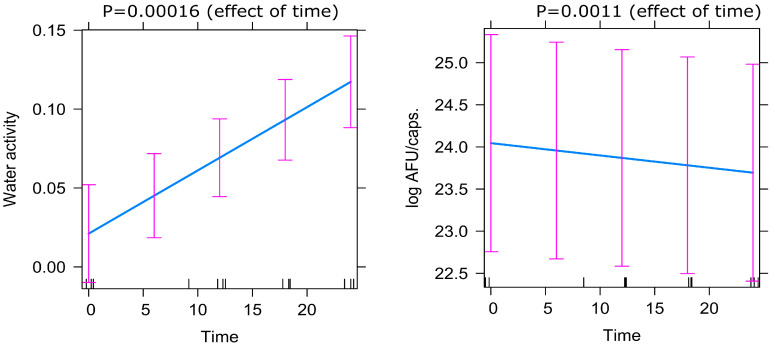
Predictor effects plot demonstrating the effect of time on Active Fluorescent Units (AFU) and water activity (a_w_) based on the mixed-effect model.

**Table 1 foods-12-04018-t001:** Parameters of microbiological quality of *Lactobacillus plantarum* 299v.

Time	0	12	24
**Microbiological Parameters**			
TAMC (total aerobic microbial count)	<10 CFU/g	<10 CFU/g	<10 CFU/g
TYMC (total yeast and molds count)	<10 CFU/g	<10 CFU/g	<10 CFU/g
Presence and count of bile-tolerant Gram-negative bacteria (*Enterobacteriaceae*)	<10 CFU/g	<10 CFU/g	<10 CFU/g
Presence of *E. coli*	Absent/1 g	Absent/1 g	Absent/1 g
Presence of *S. aureus*	Absent/1 g	Absent/1 g	Absent/1 g
Presence of *Salmonella* sp.	Absent/10 g	Absent/10 g	Absent/10 g

**Table 2 foods-12-04018-t002:** Parameters of microbiological quality of *Lactobacillus helveticus* Rosell-52 and *Bifidobacterium longum* Rosell-175.

Time	0	12	24
**Microbiological Parameters**			
TAMC (total aerobic microbial count)	<10 CFU/g	<10 CFU/g	<10 CFU/g
TYMC (total yeast and molds count)	<10 CFU/g	<10 CFU/g	<10 CFU/g
Presence and count of bile-tolerant Gram-negative bacteria (*Enterobacteriaceae*)	<10 CFU/g	<10 CFU/g	<10 CFU/g
Presence of *E. coli*	Absent/1 g	Absent/1 g	Absent/1 g
Presence of *S. aureus*	Absent/1 g	Absent/1 g	Absent/1 g
Presence of *Salmonella* sp.	Absent/10 g	Absent/10 g	Absent/10 g

**Table 3 foods-12-04018-t003:** Parameters of microbiological quality of *Bifidobacterium bifidum* Rosell^®^-71, *Lactobacillus helveticus* Rosell^®^-52, *Lactococcus lactis* Rosell^®^-1058 and *Lactobacillus casei* Rosell^®^-215.

Time	0	12	24
**Microbiological Parameters**			
TAMC (total aerobic microbial count)	<10 CFU/g	<10 CFU/g	<10 CFU/g
TYMC (total yeast and molds count)	<10 CFU/g	<10 CFU/g	<10 CFU/g
Presence and count of bile-tolerant Gram-negative bacteria (*Enterobacteriaceae*)	<10 CFU/g	<10 CFU/g	<10 CFU/g
Presence of *E. coli*	Absent/1 g	Absent/1 g	Absent/1 g
Presence of *S. aureus*	Absent/1 g	Absent/1 g	Absent/1 g
Presence of *Salmonella* sp.	Absent/10 g	Absent/10 g	Absent/10 g

## Data Availability

The datasets used and/or analyzed during the current study are available from the corresponding author on reasonable request.

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
