# Peer review of "Dynamics of Active Fluorescent Units (AFU) and Water Activity (aw) Changes in Probiotic Products—Pilot Study"

_foods, 2023, doi:10.3390/foods12214018_

Round 1

Reviewer 1 Report

Comments and Suggestions for Authors

The study offers valuable insights into the dynamic changes in active fluorescent units (AFU) and water activity (aw) in probiotic products, shedding light on their stability over time. The research highlights the significance of utilizing flow cytometry and aw measurements in evaluating probiotic product quality. However, there are several areas in which the study could be further improved or clarified:

1. P 1, L 20: When you first introduce the abbreviations "AFU" consider defining them to ensure that readers who may not be familiar with these terms can understand the context.

2. The authors  mention the use of flow cytometry (FCM) as an innovative method and briefly touch on its application in the analysis of probiotics. Consider elaborating on the existing gaps in research or limitations of traditional methods (such as the plate count method) that FCM can potentially address. This can help readers understand why the study is important.

3. P.2, L 47:   "An inherent problem of good product quality and its clinical efficacy is the counting of probiotic bacteria"  This sentence is a bit confusing and should be rephrased for clarity.

4.  P. 3, L 121:  the aim of the study is to show changes in aw and AFU levels during the aging of probiotic products. Make this objective more explicit and state it as a research question or hypothesis. For example, "This study aims to investigate how changes in water activity (aw) correlate with variations in active fluorescent units (AFU) during the aging of probiotic products."

5. Ensure consistency in units and symbols throughout the methodology. For example, check the consistency of symbols like µL and mL.

6. The study mentioned  the use of blisters composed of pharmaceutical aluminum foil/Pentapharm® Aclar® S03 for sample storage but does not delve into details regarding the quality assurance measures in place to ensure consistent and controlled storage conditions. Information on temperature monitoring, humidity control, and any potential variations in storage conditions could enhance the study's rigor.

7. The manuscript could benefit from a more comprehensive discussion of its limitations. It is essential to acknowledge potential sources of bias, confounding variables, or uncertainties in the methodology.

8. P 8, L 316: The authors state that the results are compliant with “EP” criteria, consider briefly mentioning what the EP criteria are and how your results compare to those criteria. This adds context to the compliance statement.

9. When describing the samples, the authors mentioned the strains used in each product, which is good. However, you may want to briefly explain why you chose these specific strains for the study. Did they have relevance to a particular aspect of probiotic stability?

Comments on the Quality of English Language

minor correction needed 

Reviewer 2 Report

Comments and Suggestions for Authors

The authors aimed to show that flow cytometry and water activity measurements can be used together to monitor viability of probiotic cultures in products. The use of these two parameter in relation to viability of microorganisms is not new. My expectations were that the study will show how knowledge of the one parameter, based on the correlation of results obtained when they were used to evaluate viability of probiotics, will results maybe in a quicker way of assessing the viability.  However, at the end, I still could not understand why these two methods should be sed together. There are some issues with the manuscript/study, for example, it is not known how the cultures that were put into the pharmaceutical foils were prepared, did they contain any protective agents or not, also when evaluating viability especially for products with multiple strains, when getting the AFU, do we know the counts for each of the separate strains or not? If we do, what in their methodology allows distinction of the different strains? The part were the authors evaluated the presence of different bacteria (Tables 1 to 3): It is not clear how that contributes to the study aim.  The tables (results) are also under materials and methods, which I think is the reason what this part of the study is not adequately addressed in the results and discussion. The figures have headings within the figure, which is normally done for a presentation. Over and above these, the manuscript is not well written and this makes it very difficult to follow. This could probably have contributed to my inability to grasp the message the authors intend to convey. I do not think the manuscript is suitable for publication due to lack of novelty and lack of clarity and focus.

Comments on the Quality of English Language

The authors aimed to show that flow cytometry and water activity measurements can be used together to monitor viability of probiotic cultures in products. The use of these two parameter in relation to viability of microorganisms is not new. My expectations were that the study will show how knowledge of the one parameter, based on the correlation of results obtained when they were used to evaluate viability of probiotics, will results maybe in a quicker way of assessing the viability.  However, at the end, I still could not understand why these two methods should be sed together. There are some issues with the manuscript/study, for example, it is not known how the cultures that were put into the pharmaceutical foils were prepared, did they contain any protective agents or not, also when evaluating viability especially for products with multiple strains, when getting the AFU, do we know the counts for each of the separate strains or not? If we do, what in their methodology allows distinction of the different strains? The part were the authors evaluated the presence of different bacteria (Tables 1 to 3): It is not clear how that contributes to the study aim.  The tables (results) are also under materials and methods, which I think is the reason what this part of the study is not adequately addressed in the results and discussion. The figures have headings within the figure, which is normally done for a presentation. Over and above these, the manuscript is not well written and this makes it very difficult to follow. This could probably have contributed to my inability to grasp the message the authors intend to convey. I do not think the manuscript is suitable for publication due to lack of novelty and lack of clarity and focus.

Author Response

Thank you very much for your thorough assessment of the content of my publication and the critical comments you wished to provide. They are important hints to improve the quality of my future research. I will try to address each of the comments.

First, the Reviewer points out that the methods are not new - this is true, but the combination of these two methods and their application in the food industry is new. Please bear in mind that the production of probiotic powders must be carried out under appropriate conditions and that bacterial enumeration control must be continuous. As a manufacturer of probiotics, we strive to ensure the highest quality probiotics for our consumers. We must adhere to Good Manufacturing Practice (GMP) and have a robust quality control system for probiotics that follows the standards described in the Pharmacopeia and ISO standards. The methods used in our study comply with these requirements.

Secondly, the aim of this paper was not to evaluate the proportion of live cells of each bacterial species present in the tested multistrain products, but to evaluate the number of probiotic bacteria in general during stability testing and how it relates to the water activity of these products. This practice is very commonly used by probiotic manufacturers. Also in this case both parameters are included in the products specifications and are routinely measured during finished product quality control and product stability testing. 

Thirdly, the studies presented in Table 1-3 were carried out according to the criteria of the European Pharmacopoeia for microbial purity, which is very important for the safety of food supplements. According to these criteria, the product should be free of pathogens and our tests confirm microbiological purity during stability testing.

Fourthly, the Reviewer expected the study to show how knowledge of the one parameter, based on the correlation of results obtained when it was used to assess the viability of probiotics, will lead to a faster way of assessing viability. As the title suggests, this is a pilot study. At the moment, our primary aim is to analyze more samples and to obtain the data requested by the Reviewer.

Finally, the Reviewer pointed out that it is not known how the cultures placed in the pharmaceutical foils were prepared. It is unclear whether they contained any protective agents or not.  Point 2, subsection 2.1 describes the bacterial content of the capsule used for the study. We were interested in the final product, not the intermediate stages of probiotic preparation. The methods of lyophilisation are considered confidential and, in our opinion, are not the subject of this publication.

However, in agreement with the opinion of the Reviewer quoted above, we have included the limitations of the study in the discussion, where we have presented many of the comments made in the reviews. We believe that this will enable readers to take a broader view of the issues we have.

Reviewer 3 Report

Comments and Suggestions for Authors

The manuscript by Sielatycka et al. reports about changes in aw and the number of live microbial cells (AFU) during ageing of probiotic products containing freeze-dried bacteria. Unfortunately, several improvements are needed, in terms of scientific contribution of the manuscript. Therefore, the manuscript is not acceptable for publication in Foods.

 Major concerns:

The main weakness of this manuscript is the lack of the results of identification of lactic acid bacterial strain on the species level. There are no results which determine the proportion of live cells of each bacterial species present in the tested samples containing more than one strain of lactic acid bacteria. Namely, from the data presented in this way, it can’t be concluded to which bacterial strain determined live bacterial cells belongs, and whether these strains are equally represented in the samples over a period of 24 months. For the functionality of the probiotic product it is necessary that the cells of each bacterial strains added in the probiotic preparation are alive in order to contribute to probiotic effect. Without these experimental data, the paper is not suitable for publication in the journal Foods. Without the mentioned additional experiments, the scientific contribution of this manuscript is questionable.

 Specific comments:

Abstract

Abbreviation AFU is explained on the page 4, but it is mentioned in the Abstract and in the Introduction without addition of full meaning.

2. Materials and Methods

2.1. Samples

The producer (company) of commercial samples of bacteria used in this study must be mentioned.

Figure 2

Full meaning of abbreviations TFU and FCM must be added

Figure 4

Full meaning of abbreviation AFU must be added

Author Response

Thank you for your valuable feedback. 

Response: We agree that from a scientific point of view it is highly recommended to identify bacteria at the strain level, but currently the cytometric method based on the ISO standard is not yet suitable and adopted for this purpose. The aim of this study was not to evaluate the proportion of live cells of each bacterial species present in the tested multi-strain products, but to evaluate the number of probiotic bacteria in general during stability testing and how it relates to the water activity of these products. Both parameters are included in the product specification and are routinely measured during finished product quality control and product stability testing.  For this reason, we propose flow cytometry, which is used to enumerate bacteria but not to identify individual strains. For probiotic research in industry, it is crucial to perform the study quickly and according to specific requirements, which requires adherence to established pharmacopoeial methods or ISO standards. 

We are aware of the difficulties associated with assessing the viability of probiotic bacteria in multi-strain products, which has been very well described by Wendel et al. (2022) and our purpose was not to study this very important problem. We wanted to demonstrate the relationship between these two parameters during product storage, as we believe that combining these methods offers better control of product stability than using each of them separately. We believe that the results presented by us can contribute to the development of better methods for controlling the stability of probiotics, including multi-strain probiotics. In agreement with the opinion of the Reviewer presented above, we have included and discussed the inability to identify bacterial strains as a limitation of the study.

Specific comments:

Abstract
Abbreviation AFU is explained on page 4, but it is mentioned in the Abstract and in the Introduction without addition of full meaning.

Response: The abbreviation has been added in the title and abstract.

  1. Materials and Methods

2.1. Samples

The producer (company) of commercial samples of bacteria used in this study must be mentioned. 

Response:
The final products from Sanprobi Sp. z o.o., Sp. k., (Poland) company were used in the study, which contained freeze-dried bacteria of the strains mentioned in the publication. The company name was added in point 2.1

Lactobacillus plantarum 299v - Sanprobi IBS® - capsules manufacturer—Institute Rosell-Lallemand, Montreal, Canada; LP299v strain owner-Probi AB, Lund, Sweden); Lactobacillus helveticus Rosell-52 and Bifidobacterium longum Rosell-175 - Sanprobi Stress - powder manufacturer—Institute Rosell-Lallemand, Montreal, Canada; Bifidobacterium bifidum Rosell®-71, Lactobacillus helveticus Rosell®-52, Lactococcus lactis Rosell®-1058 and Lactobacillus casei Rosell®-215 - Sanprobi 4Enteric - powder manufacturer—Institute Rosell-Lallemand, Montreal, Canada;

Figure 2
Full meaning of abbreviations TFU and FCM must be added

Figure 4
Full meaning of abbreviation AFU must be added

Response:

Full meaning of abbreviations: AFU, n-AFU, TFU and FCM were added in the caption of Figure 2, 3 and 4.

Reviewer 4 Report

Comments and Suggestions for Authors

The manuscript "Dynamics of active fluorescent units and water activity changes in probiotic products – pilot study" was well written. It presents an interesting subject with applications in the food industry. The results are consistent with the conclusions. Therefore, I present minor points to improve it:

1. Revise the language;

2. There are two tables 1;

3. Enter the number of repetitions of the experiments in the figure captions.

Comments on the Quality of English Language

 Minor editing of English language required.

Author Response

Thank you very much for your comments you wished to provide. 

Comments 1: Revise the language; 

Response 1: The article was linguistically proofread in March 2023 (MDPI certificate will be attached), however, changes were made after proofreading.

Comments 2: There are two tables 1; 

Response 2: Thank you for your comment. Corrected. The tables are numbered from 1 to 3

Comments 3: Enter the number of repetitions of the experiments in the figure captions.

Response 3: Thank you for your comment. L276: The sentence is added: The figure is representative of one production series.
The experiment was carried out for six consecutive production series.

Reviewer 5 Report

Comments and Suggestions for Authors

In this manuscript, Katarzyna Sielatycka et al. proposed a flow cytometry (FCM) combined with water activity (aw) analysis method. The method can be used to monitor the stability of bacteria number and the changes of aw and AFU levels in probiotic products containing lyophilized bacteria during aging under standard conditions. Overall, some concerns need to be addressed before further consideration. Some comments are listed as follows:

1. It is mentioned in the article that " The FSC/SSC dot plot contains all the bacteria that are separated in the next graph." Please explain which picture is specifically referred to in the next picture?

2. It is mentioned in the article that "With the increase in aw, the number of bacteria decreased but the aw at the end of the stability study did not exceed 0.15 in each of the three tested products." According to the results of Fig. 2E-F, compared with Fig. 2A-D, the bacteria in Fig. 2E-F had the most serious death. However, as can be seen from Fig. 3C, the water activity of the same bacteria was not the highest. Please explain?

3. "It is worth noting that, in each of our products, the amount of AFU decreases with the increase in aw, which never surpasses a value of 0.15 over a period of 24 months as shown in Figure 3". Please change Fig. 3 to Fig. 4.

Reviewer 6 Report

Comments and Suggestions for Authors

The article Dynamics of active fluorescent units and water activity changes in probiotic products – pilot study shows a study on the dynamics of water activity and AFU as factors to evaluate the quality of probiotics for quality control purposes.

The authors need to clarify which parameters were used to evaluate probiotics' quality because three are mentioned in the abstract: water activity (aw), FCM, and AFU. Even in the abstract, the authors need to name or define the AFU parameter, which undermines the initial understanding of the article. I suggest a reformulation of the abstract defining the AFU and a better clarification of which parameters were used to monitor the quality of probiotics.

Author Response

Thank you very much for this important comment.

Comments 1:

The article Dynamics of active fluorescent units and water activity changes in probiotic products – pilot study shows a study on the dynamics of water activity and AFU as factors to evaluate the quality of probiotics for quality control purposes.

The authors need to clarify which parameters were used to evaluate probiotics' quality because three are mentioned in the abstract: water activity (aw), FCM, and AFU. Even in the abstract, the authors need to name or define the AFU parameter, which undermines the initial understanding of the article. I suggest a reformulation of the abstract defining the AFU and a better clarification of which parameters were used to monitor the quality of probiotics.

Response 1:

I agree with the Reviewer that the information about the parameters used in the paper was unclear. Therefore, the abbreviations FCM, AFU, and aw have been expanded and the sentence describing the research aim has been corrected in abstract and manuscript. Additionally, it was explained what AFU is.

L:2; 19-24;

Round 2

Reviewer 3 Report

Comments and Suggestions for Authors

The authors included and discussed the inability to identify bacterial strains as a limitation of the study in the manuscript. Therefore, it can be accepted in its current form.